# Scoping review of evidence synthesis: Concepts, types and methods

Carla Andrea Trapé[1]*, Célia Maria Sivalli Campos[1], Cintia de Freitas Oliveira[2], João Gabriel Sanchez Tavares da Silva[3], Liza Yurie Teruya Uchimura[3], Mabel Fernandes Figueiró[3], Maritsa Carla de Bortoli[2], Sidney Marcel Domingues[4], Tatiana Yonekura[3]

1 Departamento de Enfermagem em Saúde Coletiva, Escola Enfermagem da Universidade de São de Paulo, Sao Paulo - SP, Brazil, 2 Centro de Tecnologia de Saúde, Instituto de Saúde, Sao Paulo - SP, Brazil, 3 Hcor, Sao Paulo - SP, Brazil, 4 Departamento de Doenças Raras, Biogen, Sao Paulo - SP, Brazil

* carlaens@usp.br

## Abstract

### Objectives

To systematically explore the concepts, types, and methodologies related to literature reviews and evidence synthesis.

### Methods

We conducted a scoping review using PubMed, Embase, Biblioteca Virtual da Saúde, grey literature, and websites of key international and national institutions, including the Cochrane Collaboration, Joanna Briggs Institute, Center for Reviews and Dissemination, Campbell Collaboration, and REBRATS, with searches completed through November 13, 2024. Studies were included if they identified or proposed theories and/or methodologies for evidence synthesis at international or national levels, with eligibility limited to studies published in English, Spanish, or Portuguese and no restrictions on publication year. Title and abstract screening was conducted independently by ten reviewers working in pairs, with a third reviewer resolving conflicts as needed, and full-text copies of potentially relevant articles were retrieved for further analysis.

### Results

The review included 99 studies employing a variety of evidence synthesis methods. A total of 71 terminologies for types of evidence synthesis were identified and grouped by conceptual and methodological similarities, resulting in 16 categories of evidence synthesis, each with clear differences in concepts and methods.

**Data availability statement:** All relevant data are within the paper and its Supporting Information files.

**Funding:** This review was funded by the Institutional Development Program of the Brazilian National Health System (Programa de Apoio ao Desenvolvimento Institucional do Sistema Único de Saúde - PROADI-SUS), number 25000.010649/2020-11.The funder had no role in study design, data collection and analysis, decision to publish, or preparation of the manuscript.

**Competing interests:** The authors have declared that no competing interests exist.

**Provenance and peer review:** Not commissioned; externally peer reviewed.

## Conclusions

The lack of standardization in defining and classifying review types challenges the scientific community by hindering study comparisons and appropriate methodology selection. Future research should explore the relationships between different review types and their outcomes, as well as the applicability of new methodologies across various fields.

## Introduction

In a context where scientific production is growing exponentially, literature reviews and evidence synthesis play a critical role due to their summarizing nature. These tools follow rigorous and transparent methods in their development, ensuring reliable information. As such, they are valuable for those beginning their research and anyone interested in a specific topic who does not have time to absorb the entirety of scientific literature.

As literature reviews and evidence synthesis have become established as research methods, various initiatives have been developed by international and national institutions to standardize their processes through guidelines, manuals, and standards. For instance, international organizations such as the Cochrane Collaboration and the Joanna Briggs Institute, among others, and in Brazil, initiatives by the Ministry of Health, the Brazilian Network for Health Technology Assessment (REBRATS), and the EvipNet Network promote and publish methodological guidelines for the preparation of reviews and evidence synthesis.

In recent years, mainly since 2010, there has been a significant increase in literature review studies, types of literature synthesis, and materials developed for their execution, such as checklists of mandatory items and methodological assessment tools. There is considerable diversity in the literature on this topic.

Grant and Booth [1], in his review of literature review types, highlights that "What remains largely unknown are the subtle variations in the degree of process and rigor within the different types of review." This scenario demanded the identification of a broader range of review types, opening the possibility of summarizing case studies, qualitative research, and even theoretical and conceptual results, both published and unpublished. According to Sarrami-Foroushani et al. [2], who also conducted a review on types of literature reviews, reflecting on various definitions provided in the literature, it becomes clear that, for a literature review to be effective, it must possess the following characteristics: (a) Provide strong support for a research topic and be necessary in a given field; (b) Use well-defined quality data for synthesis and analysis; (c) Select an appropriate research methodology; (d) Contribute to the development of a new scope of practice; (e) Highlight the need for further investigation into previously unexplored areas of interest.

However, while various methods for synthesizing evidence have been proposed, a comprehensive understanding of the available options and their appropriate applications is essential for researchers. This scoping review addresses this gap

by systematically exploring the concepts, types, and methodologies associated with literature reviews and evidence synthesis.

## Materials and methods

A scoping review was conducted to map theoretical/methodological studies on methods, concepts, and types of literature reviews and evidence synthesis. This approach is particularly effective for mapping literature on broad topics and in fields with limited scientific output. It also facilitates the identification of key concepts, theories, sources, and knowledge gaps by including a diverse range of studies, including theoretical, qualitative, quantitative, and review-based research [3,4].

The review followed the Preferred Reporting Items for Systematic Reviews and Meta-Analyses extension for Scoping Reviews (PRISMA-ScR) Checklist [5], published by the Joanna Briggs Institute (JBI). The protocol for this review is registered on the Zenodo. Research ethics approval was not required for this study, as it is a summary of already published literature. Patients and/or the public were not involved in the design, or conduct, or reporting, or dissemination plans of this research.

### Inclusion criteria

The review encompassed theoretical-methodological studies, literature reviews, grey literature, and guidelines that identified or proposed a theory and/or methodology for evidence synthesis at any level of scope (international and national). Only studies published in English, Spanish, or Portuguese, with no year of publication restrictions, were included. The following types of studies were included: literature reviews and guidelines that reported on types of synthesis, methodologies, and concepts.

### Exclusion criteria

Studies that used a literature review as a methodology to review a research object and did not report any method, concept, or define the type of synthesis were excluded.

### Search strategies

We searched the following electronic databases to identify published sources: PubMed, Embase, and Biblioteca Virtual da Saúde. Grey literature and websites of international and national institutions (Cochrane Collaboration, Joanna Briggs Institute, Center for Reviews and Dissemination, Campbell Collaboration, and REBRATS) that are references in methodologies for evidence synthesis, which conceptualized, analyzed, or proposed methods for conducting literature reviews and evidence synthesis were also included in the search.

The search was conducted on November 13, 2024. Studies of any design which examined checking reference list as a search method. Subject descriptors, synonyms, and keywords were used to compose the search sets in the defined information sources. The search strategies used for each database and their corresponding results can be found in S1 Appendix.

### Study selection

We examined the titles and abstracts obtained from searches of electronic databases and retrieved full-text copies of potentially relevant articles. We conducted screening for relevance using an internet-based and secured. These steps were performed using the Rayyan software by pairs of independent reviewers, with the participation of a third reviewer to resolve conflicts when necessary.

### Data extraction

Data extraction was performed by pairs, with one researcher extracting and another checking the extracted data, using a standardized data extraction form a priori. We noted information pertaining to the study design; objective of the synthesis

methodology; concept/definition of the evidence synthesis methodology addressed in the publication; proposal for an update/new methodology; steps in conducting the methodology; and steps reported in the study and their brief description (protocol development, strategy for developing and/or formulating the research question, data sources and search conduct, data extraction, methodological quality assessment and risk of bias, confidence assessment of evidence, synthesis format of findings, other methodological steps, limitations of the synthesis method).

### Critical quality appraisal

In scoping reviews, methodological quality is not assessed a priori, as they do not aim to synthesize results or identify the risk of bias [4].

### Data analysis

We summarized all relevant data from non-comparative studies descriptively, emphasizing the limitations of the study designs on the interpretations of findings. Given the theoretical nature of this scoping review, the findings were described and analyzed in terms of the concepts, types, and methods of evidence synthesis [4].

## Results

A search of the information sources yielded 41, 543 studies, and after removing duplicates, 28, 045 were screened by title and abstract. 114 studies were selected for full-text review, and of these, 99 were included. We summarized all the results of the study selection process in Fig 1 using PRISMA flow diagram.

A significant increase in the number of publications on this topic has been observed over the years. From 1995 to 2009, a total of 15 studies were published, followed by a substantial rise beginning in 2010, reaching a total of 84 studies by 2024. Regarding the country of affiliation of the first author, among the studies that reported this information, the majority originated from Brazil, followed by Canada, Australia, and the United States of America (USA).

From the 99 studies included a total of 71 distinct nomenclatures for types of evidence synthesis were identified, which were subsequently regrouped into 16 categories based on similar concepts and methods (Table 1). The detailed characteristics of these studies are provided in S2 Appendix. Excluded studies and their reasons are listed in S3 Appendix.

Inventory of references [7,8], summary of abstracts [7], rapid reviews [1,7–18] and, technical-scientific report [7,19] represent efficient strategies for synthesizing evidence in time-constrained contexts. While reference inventory primarily focus on cataloging relevant literature [7,8], summary of abstracts, rapid reviews and technical-scientific report employ streamlined methodologies to expedite the synthesis process [20,21]. These methods are best suited for providing an initial overview of the evidence landscape, particularly in scenarios like public health emergencies and policy formulation.

Inventory of references refers to a compilation of existing literature without formal synthesis or critical appraisal, examples include the list of references [7,8]. A key characteristic is the absence of any analysis or interpretation of the included studies. They might be simple lists of references or more structured databases with basic bibliography on specific information [7,8]. Summary of abstracts often involve theme categorization based only from the abstracts of included scientific studies, providing a structured approach to identifying key findings [7]. For example a table summarizing the primary outcomes reported in abstracts of clinical trials investigating specific drug.

Rapid reviews utilize various techniques, such as narrowing the scope of the research question, limiting languages, and selectively targeting specific databases, to accelerate the evidence synthesis process [1,7–16,18,20,21]. They aim to provide a more comprehensive synthesis than a summary of abstracts, but may still compromise comprehensiveness and depth compared to full systematic reviews.

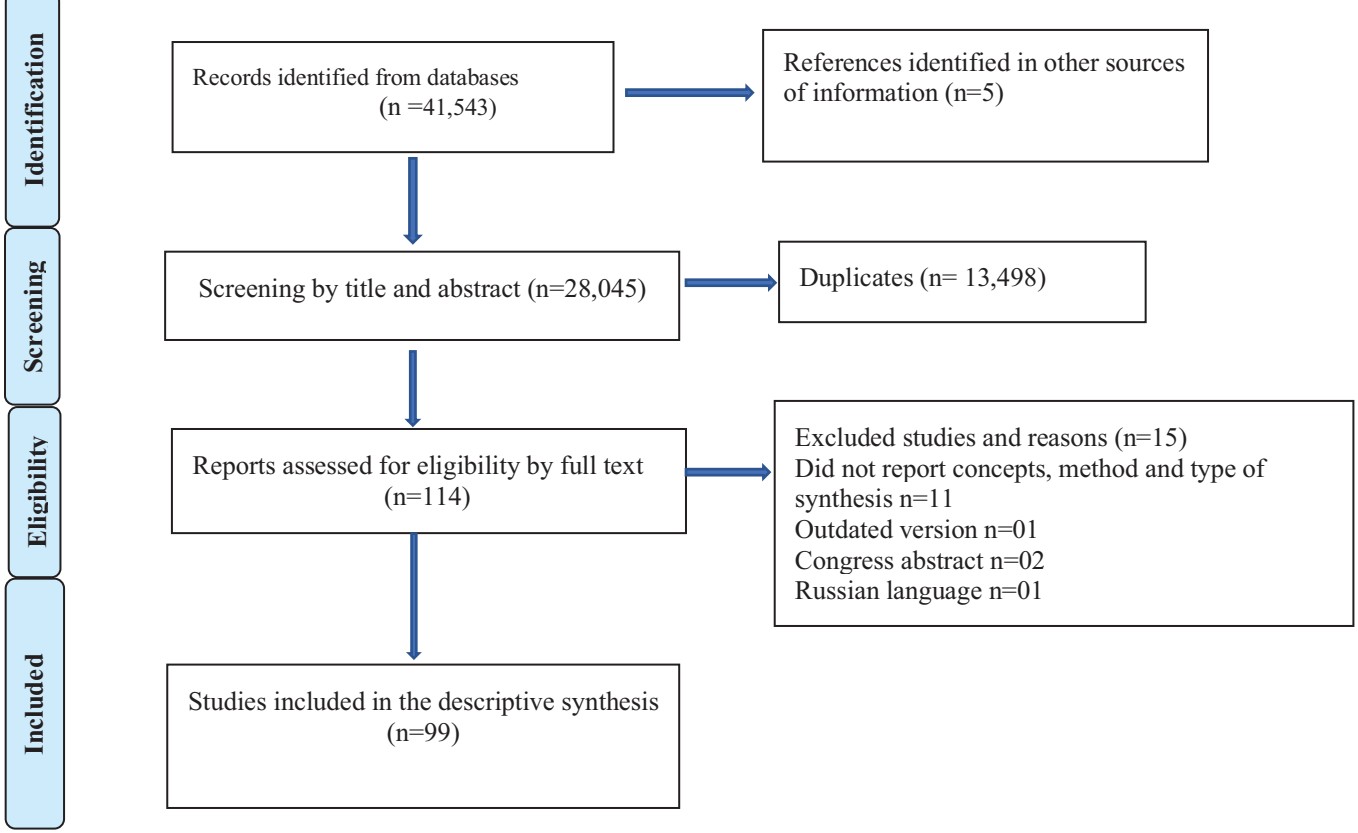

Adapted from: Page MJ, McKenzie JE, Bossuyt PM, Boutron I, Hoffmann TC, Mulrow CD, et al. The PRISMA 2020 statement: an updated guideline for reporting systematic reviews. BMJ 2021;372:n71. doi: 10.1136/bmj.n71

**Fig 1. Systematic review flow chart of records identification and study screening and selection.** Adapted from: Page et al. [6].

A technical-scientific report is a rapid synthesis that integrates economic, regulatory, and health technology assessment (HTA) information to support decision-making. It provides timely evidence to inform policy and practice decisions on health technologies [7,19]. A defining feature is their focus on providing actionable recommendations, for example a report evaluating the cost-effectiveness and safety of a new medical device for a national health system.

Evidence maps, also referred to as systematic maps, are structured or graphical representations of available research within a specific field. These methods prioritize categorization and visualization over synthesis, focusing on mapping the distribution of studies rather than critically appraising or integrating their findings [1,13,22–26]. Among the various evidence synthesis methods, evidence maps are the only type that offer a unique framework for visually representing the evidence landscape within a specific sector or topic area [1,13,22–27]. They are typically structured in a matrix format, with columns and rows outlining key interventions, outcomes, and study characteristics. As a knowledge translation tool, they support decision-making by highlighting both effective and ineffective interventions, while also pointing to areas that need further high-quality research [1,13,22–26].

A scoping review also serves to map the available evidence but follow a more rigorous methodology and a detailed analysis of study results. Unlike evidence maps, scoping reviews systematically compile data from individual studies,

**Table 1. Types of evidence synthesis identified grouped according to the similarity of concept and method.**

| | |
|---|---|
| 1. **Inventory of references**<br>Inventory of evidence<br>List of references<br>2. **Rapid review**<br>Rapid Appraisal of Evidence<br>Rapid response<br>Rapid assessment review<br>3. **Summary of abstracts**<br>Evidence summary<br>4. **Evidence map**<br>Systematic map<br>5. **Technical-scientific report**<br>6. **Critical review**<br>7. **Scoping review**<br>Scoping meta-review<br>Mapping review<br>8. **Mixed methods review**<br>Mixed-methods review<br>Systematic mixed-methods review<br>9. **Reviews of economic evaluations**<br>Complete economic assessment<br>10. **Review of systematic reviews**<br>Overview<br>Overview of systematic methods<br>Umbrella review<br>Meta-review<br>11. **Text and opinion review**<br>Expert opinion review<br>Opinion review<br>Policy review<br>Systematic review of text and opinion<br>12. **Integrative Review**<br>Aggregative review | 13. **Narrative review**<br>Traditional review<br>Literature review<br>Bibliographic research<br>Narrative literature review<br>State-of-the-art review<br>Expository review<br>Questioning review<br>Historical review<br>Configurative review<br>Meta-narrative review<br>Concept synthesis<br>14. **Realist review**<br>Realist synthesis<br>15. **Quantitative systematic review**<br>Systematic review with meta-analysis<br>Systematic review with meta-analysis of randomized clinical trials<br>Systematic review of interventions<br>Systematic review and meta-analysis of diagnostic accuracy studies<br>Systematic review with meta-analysis of studies on risk factors and prognosis<br>Systematic review of diagnostic test accuracy<br>Systematic review of prevalence/incidence<br>Efficacy review<br>Prognostic review<br>**Qualitative**<br>Systematic review of qualitative evidence<br>Review of experimental evaluations<br>**Others**<br>Synthesis of living evidence<br>Test review<br>Meta-summary review<br>Living systematic review<br>Systematic literature review<br>Systematized review<br>Review of methodologies types<br>Psychometric review<br>16. **Evidence synthesis for policy**<br>Policy briefs |

Source: Authors.

providing a comprehensive overview that helps identify specific topics and knowledge gaps that require further investigation [1–3,7,13,15,27–39]. They help identify key concepts, research trends, and knowledge gaps that require further investigation.

Although they lay a foundation for future investigations, they do not assess the risk of bias in the included evidence and therefore cannot guarantee evidence quality or support clinical recommendations based on their findings [1–3,7,13,15,27–30,32,34–40]. Scoping reviews often use the Population, Concept, and Context (PCC) acronym to define the scope of the review and structure the analysis [25].

Critical reviews offer in-depth analyses of existing scientific literature on a specific topic. They employ predefined criteria to evaluate strengths and weaknesses of the available evidence, aiming to identify research gaps, formulate hypotheses or propose new theoretical models. A defining characteristic is the emphasis on critical reading, data extraction, and structured reporting, rather than adherence to explicit search, synthesis, or analysis methods [1,5,13,37,39]. Narrative reviews, similarly, synthesize information from existing literature, but take a broader perspective and may not employ a rigorous systematic methodology [1,5,13,22].

While valuable, both critical [1,5,13,37,39] and narrative reviews [1,5,13,22,37,39,41–47] often lack the methodological rigor of systematic reviews. The absence of explicit methodological guidelines can lead to inconsistencies in the selection and interpretation of evidence, making it difficult to assess the overall quality of the review. This limitation can hinder the reproducibility of findings and make it challenging to draw definitive conclusions based on the evidence presented.

A review of systematic reviews (also known as an "umbrella review") synthesizes multiple systematic reviews to provide a comprehensive examination of interventions across different conditions, outcomes, or settings. This approach is particularly useful for well-studied topics, offering insights into common challenges and existing guidelines [1,13–15,32,33,37,40,48–51]. Integrative reviews, in contrast, synthesize evidence from diverse sources, including both qualitative and quantitative studies, to provide a holistic understanding of a phenomenon. They are particularly useful for exploring complex concepts and identifying knowledge gaps [5,13,22,40,46,47,52–57]. The methodologies of systematic, integrative, mixed-methods, quantitative, and qualitative reviews, while underpinned by a shared commitment to rigor, are adaptable to the specific demands of different research questions. While all systematic reviews adhere to rigorous methodological principles, they diverge in their approaches to data synthesis [1,13–15,28,33,37,40,48–51]. Mixed methods systematic reviews specifically combine quantitative and qualitative data to offer a more comprehensive understanding [1,5,14,22,40,46,57–61]. On the other hand, quantitative systematic reviews focus exclusively on numerical data, aiming to quantify the magnitude of effects providing structured methodologies to evaluate prevalence, diagnostic accuracy, and intervention effectiveness [1,15,19,27,40,41,43,46,47,60,62–84]. Conversely, qualitative systematic reviews delve into the subjective experiences and perspectives captured in textual data. These distinct approaches, while complementary, serve different purposes and contribute to a more nuanced understanding of research topics [1,14,34,46,67,85].

Several specialized review types further refine the evidence synthesis landscape. In economic evaluation reviews, evidence synthesis is conducted to determine the most accurate estimates of health outcomes while considering both costs and health impacts. When analyzing intervention prioritization, budgetary implications, and resource allocation, the performance of a specific technology is a key factor [7,67,86].

Text and opinion reviews focus on the qualitative analysis of texts, documents, and reports to understand the context and generate theories [42,67,87,88]. They delve into the nuances of language and discourse to explore perspectives and interpretations. Realist reviews seek to identify the mechanisms and contexts through which interventions work, combining qualitative and quantitative analysis [5,37,39,89,90]. For example, a review examining the implementation of a school-based health promotion program, exploring how the program works in different school settings and for different student populations. Living systematic reviews, apart from other types of reviews, is their continuous updating process, integrating new evidence as it becomes available to ensure that findings remain current and relevant [91].

Finally, evidence synthesis for policy aims to synthesize the best available evidence to inform decision-making, using both quantitative and qualitative data [7,19,92]. Although each approach has its specific focus, they complement each other and can be used in combination to gain a more comprehensive understanding of a given topic. The choice of the most appropriate approach will depend on the research objective and research questions.

## Discussion

This scoping review expands upon existing typologies of evidence synthesis types by including a broader range of review and synthesis types beyond traditional literature reviews. Scoping reviews are particularly valuable in areas with limited scientific output, as they provide a comprehensive overview of the available evidence and identify knowledge gaps.

The analysis of 94 studies revealed the diversity of evidence synthesis methods, with no single study encompassing all identified types. This highlights the need for a comprehensive typology to guide researchers in selecting appropriate methods. We described the concept and steps reported by the authors in the studies, providing researchers and anyone interested in the topic with an overview of what each type of review and synthesis entails and its applicability.

Among all the methods, rapid synthesis is a time-efficient approach to evidence synthesis that employs strategies to reduce the execution time across various stages, such as study identification and methodological quality assessment. However, the specific definition and timeline for rapid synthesis can vary, as evidenced by the diverse terminology used in the literature.

In summary, this review identified 74 terminologies for types of evidence synthesis, which were grouped by similarity of concept and method, resulting in 16 categories of evidence synthesis types due to clear differences in concepts and methods for each one. Categorization is only useful when supported by sufficient consensus or official guidance to remove ambiguity in methods and processes. In 2009, the typology published by Grant and Booth, with 14 types of reviews, highlighted "frequent inconsistencies or overlaps between the descriptions of nominally different review types" [1]. We observed the same in our scoping review.

In 2012, Gough et al. provided clarifications between different review designs and methods but did not offer a taxonomy of review types. The rationale for this was that, in the field of evidence synthesis, "the rate of development of new review approaches is too fast and the overlap of approaches is too great for this to be useful" [22].

According to Munn et al. [67], the application of evidence can be significantly hindered when it does not align with the situations professionals (or guideline developers) face. Therefore, selecting an appropriate review type that addresses relevant clinical and policy questions is essential.

Various factors may prompt an author to conduct a review, with or without a clearly defined clinical or policy question from the outset. Understanding the different review types and the questions they are designed to answer is important to a review's success [67].

Selecting the appropriate review type poses a challenge for researchers due to the wide variety of available methodologies. The methodological strengths and limitations of each approach depend largely on the research objectives. When a rapid response to urgent question is required, an Inventory of references or a Summary of abstracts may be appropriate. However, these methods often compromise the scope and depth of analysis. To mitigate this limitation, a rapid review can be conducted, as it employs strategies to streamline the search process while maintaining a structured quality assessment of included studies.

For researchers seeking a broader scope with greater methodological rigor, Evidence maps, Scoping reviews or mixed methods reviews, may be suitable. However these approaches typically require more time, particularly if a formal quality assessment of included studies is necessary, as in the case of Systematic review and Reviews of Systematic reviews.

In qualitative research, when the objective is to contextualize a topic, develop theoretical constructs, or aggregate predefined concepts without a strict requirement for explicit synthesis and analytical methods, Critical Reviews, Narrative Reviews, or Integrative Reviews may be appropriate.

Certain research questions necessitate specific review methodologies. For instance, when conducting a technological assessment of a drug, technique, or medical device in conjunction with economic evaluations to determine cost-effectiveness, a Technical-Scientific report may be the most suitable approach. If a more rigorous methodological framework is required for economic analyses, an Economic Evaluation review should be considered. When analyzing interventions or programs that involve multiple interacting factors, a Realist review is particularly valuable. Additionally, if the research aims to complement empirical evidence with expert insights and professional experience, a Text and Opinion review can be utilized.

Further details on the characteristics, appropriate applications, methodological stages, data extraction and analysis and available guidelines for using synthesis and/or examples of synthesis can be found in S2 Appendix.

Thus, decision-making informed by scientific evidence requires studies with a clear definition of the type of evidence synthesis and the steps to be developed, in accordance with the time and resources available.

Given the exploratory nature of this study, the search for published literature was constrained by language restrictions and database limitations, potentially limiting the overall comprehensiveness of the findings.

## Conclusions

The lack of standardization in the definition and classification of review types poses a significant challenge for the scientific community. Our analysis revealed a wide variety of terminologies and methodologies, which can hinder comparisons between studies and make it difficult to select the most appropriate review type. To address this issue, we propose the development of standardized guidelines that establish clear, universally accepted criteria for defining and classifying review types. These guidelines should outline key characteristics, methodologies, and appropriate use cases for each review type, ensuring greater consistency in research practices. Additionally, we encourage the scientific community to collaborate in creating a consensus-based glossary of terms and definitions for different types of reviews.

Also, we propose the development of a structured framework for defining and classifying review types, based on: standardization, applicability, and adaptability. Standardization involves creating clear, universally accepted criteria that define the scope, methodology, and expected outcomes of each review type. Applicability ensures that the classification system is practical and useful for researchers, healthcare professionals, and policymakers. Adaptability recognizes that the field is constantly evolving and requires a flexible structure that can integrate new methodologies as they emerge.

A clear and consistent classification is essential to assist healthcare professionals and decision-makers in identifying the most relevant evidence for their needs. Investing in training programs for researchers and healthcare professionals is important to ensure they understand the nuances and applications of different review types. Future research could explore the relationship between different review types and their outcomes, as well as investigating the applicability of these methodologies in various fields.

## Supporting information

**S1 Appendix. Search strategies.**
(DOCX)

**S2 Appendix. Characteristics for each method.**
(DOCX)

**S3 Appendix. Excluded.**
(DOCX)

## Acknowledgments

With thanks to Alanis Amorim Angotti, Bruna Carolina de Araujo and Denila Bueno Silva for their assistance in screening of literature and data extraction.

## Author contributions

**Conceptualization:** Liza Yurie Teruya Uchimura, Mabel Fernandes Figueiró, Tatiana Yonekura.

**Data curation:** Carla Andrea Trapé, Célia Maria Sivalli Campos, Cintia de Freitas Oliveira, João Gabriel Sanchez Tavares da Silva, Liza Yurie Teruya Uchimura, Mabel Fernandes Figueiró, Maritsa Carla de Bortoli, Sidney Marcel Domingues, Tatiana Yonekura.

**Formal analysis:** Liza Yurie Teruya Uchimura, Mabel Fernandes Figueiró, Tatiana Yonekura.

**Investigation:** Célia Maria Sivalli Campos, Cintia de Freitas Oliveira, João Gabriel Sanchez Tavares da Silva, Liza Yurie Teruya Uchimura, Mabel Fernandes Figueiró, Maritsa Carla de Bortoli, Sidney Marcel Domingues, Tatiana Yonekura.

**Methodology:** Liza Yurie Teruya Uchimura, Mabel Fernandes Figueiró, Maritsa Carla de Bortoli, Tatiana Yonekura.

**Project administration:** Liza Yurie Teruya Uchimura, Mabel Fernandes Figueiró, Tatiana Yonekura.

**Supervision:** Liza Yurie Teruya Uchimura, Tatiana Yonekura.

**Validation:** Liza Yurie Teruya Uchimura, Mabel Fernandes Figueiró, Tatiana Yonekura.

**Writing – original draft:** Carla Andrea Trapé, Célia Maria Sivalli Campos, Cintia de Freitas Oliveira, João Gabriel Sanchez Tavares da Silva, Liza Yurie Teruya Uchimura, Mabel Fernandes Figueiró, Maritsa Carla de Bortoli, Sidney Marcel Domingues, Tatiana Yonekura.

**Writing – review & editing:** Carla Andrea Trapé, Célia Maria Sivalli Campos, Cintia de Freitas Oliveira, João Gabriel Sanchez Tavares da Silva, Liza Yurie Teruya Uchimura, Mabel Fernandes Figueiró, Maritsa Carla de Bortoli, Sidney Marcel Domingues, Tatiana Yonekura.

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
