## [Decision Letter · Decision Letter 0]

17 Jan 2025

PONE-D-24-56232Scoping Review of Evidence Synthesis: Concepts, Types and MethodsPLOS ONE

Dear Dr. Trapé,

Thank you for submitting your manuscript to PLOS ONE. After careful consideration, we feel that it has merit but does not fully meet PLOS ONE’s publication criteria as it currently stands. Therefore, we invite you to submit a revised version of the manuscript that addresses the points raised during the review process.

We look forward to receiving your revised manuscript.

Kind regards,

André Ramalho, PhD

Academic Editor

PLOS ONE

Journal Requirements:

https://bmcmedresmethodol.biomedcentral.com/articles/10.1186/s12874-017-0468-4?

In your revision ensure you cite all your sources (including your own works), and quote or rephrase any duplicated text outside the methods section. Further consideration is dependent on these concerns being addressed.

4. We note that Figure 2 in your submission contain [map/satellite] images which may be copyrighted. All PLOS content is published under the Creative Commons Attribution License (CC BY 4.0), which means that the manuscript, images, and Supporting Information files will be freely available online, and any third party is permitted to access, download, copy, distribute, and use these materials in any way, even commercially, with proper attribution. For these reasons, we cannot publish previously copyrighted maps or satellite images created using proprietary data, such as Google software (Google Maps, Street View, and Earth). For more information, see our copyright guidelines: http://journals.plos.org/plosone/s/licenses-and-copyright.

5. Please remove all personal information, ensure that the data shared are in accordance with participant consent, and re-upload a fully anonymized data set. 

Reviewers' comments:

Reviewer's Responses to Questions

**Comments to the Author**

1. Is the manuscript technically sound, and do the data support the conclusions?

Reviewer #1: Yes

Reviewer #2: Yes

2. Has the statistical analysis been performed appropriately and rigorously? 

Reviewer #1: N/A

Reviewer #2: N/A

3. Have the authors made all data underlying the findings in their manuscript fully available?

Reviewer #1: Yes

Reviewer #2: Yes

4. Is the manuscript presented in an intelligible fashion and written in standard English?

Reviewer #1: Yes

Reviewer #2: Yes

5. Review Comments to the Author

Reviewer #1: With my understanding and knowledge, the manuscript appears to be technically good. A well-structured scoping review strategy is used to investigate various evidence synthesis methodologies carefully. The methodology follows existing scoping review procedures, such as the PRISMA-ScR checklist, which is widely accepted in the area.

Also, the authors have stated that all data underlying the findings described in their manuscript are fully available without restriction. This is explicitly mentioned in their Data Availability Statement, which indicates that all relevant data are included within the manuscript and its supporting information files.

Some insights might need to be addressed for more clarification of the reader:

1. The categorisation of evidence synthesis types is insightful, though further clarity and distinction between categories could enhance the manuscript's utility. The manuscript identifies 71 terminologies and groups them into 16 categories, but some categories lack clear distinctions, leading to potential confusion.

Provide more precise definitions and examples for each category to differentiate between them clearly. Consider adding specific criteria or characteristics that define each type.

2. While the manuscript lists various types of reviews, it does not provide sufficient in-depth analysis of their methodological strengths and weaknesses.

Expand on the methodological critique of each review type, discussing their specific advantages, limitations, and best use cases. This will offer readers a deeper understanding of when and how to apply each methodology.

3. The conclusion highlights the need for standardization but does not provide concrete suggestions for achieving this.

Include more detailed recommendations or a proposed framework for standardizing the terminology and methodologies. This could involve suggesting guidelines or collaboration with key stakeholders in the field.

4. Need to ensure consistent use of terminology throughout the manuscript to avoid confusion, especially when referring to the different types of evidence synthesis. Some terms like "rapid review," "systematic map," "scoping review," "narrative review," "realist review," and "integrative review" need to be clearly defined and consistently used throughout the manuscript. Need more precise definitions to avoid overlap and ambiguity.

Reviewer #2: - In a world where scientific production is exponentially growing, literature reviews and evidence synthesis play a critical role, as authors mentioned. This manuscript proves to be interesting and relevant in theme and scope considering it analyzes variations in the degree of process and rigor within the different types of review, thus aiming to explore the concepts, types, and methodologies associated with literature reviews and evidence synthesis.

Nevertheless, it is important to notice some aspects. Regarding Methods section, authors mention “Subject descriptors, synonyms, and keywords were used to compose the search sets in the defined information sources”. It would be interesting to understand whether authors did calibrate the query through a sensitivity analysis of the used terms instead of simply identifying and joining keywords and index terms in a search expression. Besides, in Study selection section, authors “conducted screening for relevance” using “Rayyan software by pairs of independent reviewers with the participation of a third reviewer to resolve conflicts when necessary”. It would be interesting to know de extent of agreement between investigators in these phase (for example, by calculating the percentage calculation of agreement or Cohen’s kappa coefficient (which is a more robust statistical measure of agreement)).

- By registering the protocol of this study, designing it in accordance with the Preferred Reporting Items for Systematic Reviews and Meta-Analyses extension for Scoping Reviews (PRISMA-ScR) Checklist, authors present a feasible methodology which allows the work replicability, thus leading to a meaningful outcome.

Given all these aspects, it is believed this manuscript is suitable for publication.

Best regards.

6. PLOS authors have the option to publish the peer review history of their article (what does this mean? ). If published, this will include your full peer review and any attached files.

**Do you want your identity to be public for this peer review?** For information about this choice, including consent withdrawal, please see our Privacy Policy .

Reviewer #1: No

Reviewer #2: **Yes: ** Bruno Filipe Coelho da Costa

---

## [Author Response · Author response to Decision Letter 1]

12 Mar 2025

Dear Editorial Board,

We sincerely appreciate the time and effort that the editorial team and reviewers have dedicated to evaluating our manuscript. We have carefully addressed each of the comments and suggestions provided, making the necessary revisions to ensure our manuscript meets PLOS ONE’s standards. Below, we provide detailed responses to each concern raised.

1. Manuscript Formatting:

We have revised our manuscript to comply with PLOS ONE's style requirements, following the provided formatting guidelines for the main text and title, authors, and affiliations.

2. Text Overlap with Previous Publications:

We have thoroughly reviewed the manuscript for any overlapping text with the publication referenced (https://bmcmedresmethodol.biomedcentral.com/articles/10.1186/s12874-017-0468-4). Any duplicated text outside the methods section has been properly cited, rephrased, or removed as necessary to avoid redundancy.

3. Grant Information Consistency:

We have corrected and aligned the grant information across the ‘Funding Information’ and ‘Financial Disclosure’ sections, ensuring accurate and consistent details, including grant numbers.

4. Copyrighted Figures (Figure 2):

We have either removed Figure 2 with an alternative that is compliant with the CC BY 4.0 license.

5. Data Anonymization:

All personal information has been removed from the dataset to comply with PLOS ONE’s data-sharing policy. We have re-uploaded a fully anonymized dataset, ensuring that no hidden columns contain sensitive information.

6. Supporting Information Captions:

Captions for all Supporting Information files have been added at the end of the manuscript, with corresponding in-text citations updated accordingly.

7. Reference List Review:

We have reviewed our reference list to ensure completeness and accuracy.

Responses to Reviewer Comments:

1. Definitions and Examples for Review Types:

○ We have refined and expanded the definitions of different review types, providing clearer distinctions and specific examples to enhance clarity. To address this, we have refined and expanded our definitions, ensuring that each review type is more precisely delineated. Additionally, we have provided concrete examples to illustrate these distinctions, which are included in the appendix table for reference.

○ These revisions aim to enhance clarity and assist readers in understanding the methodological approaches associated with each review type.

2. Methodological Strengths and Weaknesses of Review Types:

○ We appreciate the reviewer’s suggestion to provide a more in-depth analysis of the methodological strengths and weaknesses of each review type. In response, we have incorporated a discussion section (lines 159-166, 168-174, 177-184, 187-191, 196, 197, 201-2012, 217-224, 231-233, 237, 243-248, 292-316) detailed critique that outlines the specific advantages, limitations, and best use cases for each type of review. This additional discussion enhances the manuscript’s methodological rigor and provides clearer guidance for researchers on selecting the most appropriate review methodology for their objectives.

○ To maintain clarity and accessibility, we have compiled these methodological considerations into an appendix table, where each review type is systematically analyzed. We believe this format allows readers to quickly compare and contrast different methodologies while ensuring the main text remainsponde focused and concise.

3. Standardization of Terminology:

○ The conclusion now includes concrete suggestions for standardizing terminology and methodologies, including a proposed framework and collaboration strategies with key stakeholders (lines 334-344).

4. Terminology Consistency:

○ We have ensured that terms such as "rapid review," "systematic map," "scoping review," "narrative review," "realist review," and "integrative review" are clearly defined and consistently used throughout in the results section.

5. Study Selection Process and Inter-Rater Agreement:

○ We appreciate the reviewer´s suggestion regarding the calibration of search queries and the assessment of inter-rater agreement during the study selection process. Regarding the search strategy and study selection process, our team conducted multiple calibration exercises among reviewers before the formal screening process began. These calibration steps ensured a shared understanding of inclusion and exclusion criteria, reducing the likelihood of discrepancies during the screening phase. Given this iterative alignment, we did not find it necessary to apply additional agreement measures such as Cohen’s kappa coefficient. However, we acknowledge the importance of such metrics and have clarified our methodological approach in the manuscript.

We sincerely appreciate the reviewers' constructive feedback, which has helped us strengthen our manuscript. We hope that our revisions adequately address all concerns and that the manuscript is now suitable for publication in PLOS ONE.

Thank you for your time and consideration. We look forward to your response.

Best regards,

Dra. Carla Andrea Trapé

---

## [Decision Letter · Decision Letter 1]

10 Apr 2025

Scoping review of evidence synthesis: concepts, types and methods

PONE-D-24-56232R1

Dear Dr. Trapé,

We’re pleased to inform you that your manuscript has been judged scientifically suitable for publication and will be formally accepted for publication once it meets all outstanding technical requirements.

Kind regards,

André Luis C Ramalho, PhD

Academic Editor

PLOS ONE

Additional Editor Comments:

The peer review process for this manuscript was conducted in accordance with PLOS One’s editorial policies, involving two independent reviewers. In the first round, one reviewer recommended acceptance, while the other suggested minor revisions. The authors addressed all comments with significant improvements, and upon reevaluation, the second reviewer updated their recommendation to acceptance. A third reviewer was provisionally invited in case of continued divergence; however, this step proved unnecessary, and the invitation was withdrawn. With both reviewers now in agreement and the peer review process duly fulfilled, I, as editor, recommend the manuscript for acceptance and publication.

Reviewers' comments:

Reviewer's Responses to Questions

**Comments to the Author**

1. If the authors have adequately addressed your comments raised in a previous round of review and you feel that this manuscript is now acceptable for publication, you may indicate that here to bypass the “Comments to the Author” section, enter your conflict of interest statement in the “Confidential to Editor” section, and submit your "Accept" recommendation.

Reviewer #1: All comments have been addressed

2. Is the manuscript technically sound, and do the data support the conclusions?

Reviewer #1: Yes

3. Has the statistical analysis been performed appropriately and rigorously? 

Reviewer #1: N/A

4. Have the authors made all data underlying the findings in their manuscript fully available?

Reviewer #1: (No Response)

5. Is the manuscript presented in an intelligible fashion and written in standard English?

Reviewer #1: Yes

6. Review Comments to the Author

Reviewer #1: Required a basic proof reading. (Example: line 67 need to be written more clearly). The submission looks over wordy.

Author's have addresssed the previous comments.

7. PLOS authors have the option to publish the peer review history of their article (what does this mean? ). If published, this will include your full peer review and any attached files.

**Do you want your identity to be public for this peer review?** For information about this choice, including consent withdrawal, please see our Privacy Policy .

Reviewer #1: **Yes: ** Rahul Kumar Jha

---

## [Editor Report · Acceptance letter]

PONE-D-24-56232R1

PLOS ONE

Dear Dr. Trapé,

I'm pleased to inform you that your manuscript has been deemed suitable for publication in PLOS ONE. Congratulations! Your manuscript is now being handed over to our production team.

Kind regards,

on behalf of

Prof. Dr. André Luis C Ramalho

Academic Editor

PLOS ONE